# Development of Molecular Markers for Predicting Radish (*Raphanus sativus*) Flesh Color Based on Polymorphisms in the *RsTT8* Gene

**DOI:** 10.3390/plants10071386

**Published:** 2021-07-06

**Authors:** Soyun Kim, Keunho Yun, Han Yong Park, Ju Young Ahn, Ju Yeon Yang, Hayoung Song, O New Lee, Yoonkang Hur, Man-Ho Oh

**Affiliations:** 1Department of Biological Sciences, Chungnam National University, Daejeon 34134, Korea; soyun9096@naver.com (S.K.); keunho0307@gmail.com (K.Y.); wnduds357@naver.com (J.Y.A.); sjss508@naver.com (J.Y.Y.); hayoung282@hanmail.net (H.S.); 2College of Life Sciences, Sejong University, Seoul 05006, Korea; hypark@sejong.ac.kr (H.Y.P.); onewlee@sejong.ac.kr (O.N.L.)

**Keywords:** *RsTT8*, InDels, SNPs, flesh color, molecular marker

## Abstract

Red radish (*Raphanus sativus* L.) cultivars are a rich source of health-promoting anthocyanins and are considered a potential source of natural colorants used in the cosmetic industry. However, the development of red radish cultivars via conventional breeding is very difficult, given the unusual inheritance of the anthocyanin accumulation trait in radishes. Therefore, molecular markers linked with radish color are needed to facilitate radish breeding. Here, we characterized the *RsTT8* gene isolated from four radish genotypes with different skin and flesh colors. Sequence analysis of *RsTT8* revealed a large number of polymorphisms, including insertion/deletions (InDels), single nucleotide polymorphisms (SNPs), and simple sequence repeats (SSRs), between the red-fleshed and white-fleshed radish cultivars. To develop molecular markers on the basis of these polymorphisms for discriminating between radish genotypes with different colored flesh tissues, we designed four primer sets specific to the *RsTT8* promoter, InDel, SSR, and WD40/acidic domain (WD/AD), and tested these primers on a diverse collection of radish lines. Except for the SSR-specific primer set, all primer sets successfully discriminated between red-fleshed and white-fleshed radish lines. Thus, we developed three molecular markers that can be efficiently used for breeding red-fleshed radish cultivars.

## 1. Introduction

Radish (*Raphanus sativus* L.; Brassicaceae) is an economically important root vegetable cultivated worldwide for its edible taproots, sprouts, and seed oil. Taproots of radish contain dietary fiber, carbohydrates, and essential minerals and nutrients that benefit human health [1]. In addition, red radishes contain significant amounts of anthocyanins, phenolics, and flavonoids, which contribute substantially to their antioxidant activity [1,2,3]. Radish taproots typically display four types of color patterns: red skin with red flesh (RsRf), green skin with red flesh (GsRf), red skin with white flesh (RsWf), and white skin with white flesh (WsWf).

One of the radish breeding targets over the past three decades has been the development of red radish cultivars with improved functionality [4,5]. Red radish cultivars are a potential source of natural colorants, as they are rich in anthocyanins, which are highly stable and highly similar to the artificial pigment, Food Red No. 40 [6,7]. Among the six types of anthocyanins, pelargonidin-based anthocyanins are mainly found in red radishes, which enable their differentiation from other radishes containing cyanidin- or delphinidin-based anthocyanins [1,8,9,10,11].

Anthocyanin is synthesized via the flavonoid pathway by two groups of genes: early biosynthesis genes (EBGs) and late biosynthesis genes (LBGs)[12,13]. The EBG group consists of chalcone synthase (*CHS*), chalcone isomerase (*CHI*), flavanone 3-hydroxylase (*F3H*), flavanone 3’-hydroxylase (*F3’H*), and flavonol synthase (*FLS*) [14,15,16,17]; whereas, the LBGs include dihydroflavonol 4-reductase (*DFR*), leucoanthocyanidin oxygenase/anthocyanidin synthase (*LDOX/ANS*), anthocyanidin reductase (*ANR*), and UDP-glucose: flavonoid 3-*O*-glucosyltransferase (*UD3GT*), which are activated by the MYB-bHLH-WD40 (MBW) complex [12,17,18,19]. The MBW complex is composed of an R2R3-type MYB transcription factor MYB123 (encoded by *TRANSPARENT TESTA 2* [*TT2*]), a basic helix-loop-helix (bHLH) protein bHLH042 (encoded by *TT8*), and a WD40 repeat (WDR)-containing protein (encoded by *TRANSPARENT TESTA GLABRA 1* [*TTG1*]).

The bHLH proteins comprise a large class of transcription factors divided into 26 subgroups [20]. Flavonoid-related bHLH proteins, which cluster into subgroup IIIf, contain an MYB-interaction region (MIR) at the N-terminus, followed by a WDR-interacting region (WD), an acidic domain (AD), a bHLH domain, and the C-terminal region; which together mediate the homo- or heterodimerization of bHLH proteins [21,22]. Thus, bHLH transcription factors, together with R2R3 MYB proteins, play a crucial role in anthocyanin biosynthesis in plants.

Several flavonoid-related *bHLH* genes have been identified in plants: *GLABRA3* (*GL3*) [23,24,25,26,27,28], *ENHANCER OF GLABRA3* (*EGL3*) [24,25,26], *MYC1* [29,30], and *TT8* [25,28,31,32]. Each bHLH gene is involved in the regulation of two or more traits in a partially redundant manner. The *RsTT8* gene was recently isolated and characterized [33], and its role in anthocyanin accumulation is supported by genomic and transcriptomic studies. The defective expression of *TT8* caused by either transposable element (TE) insertion or mutation has been shown to affect anthocyanin accumulation in *Brassica* species, including *B. rapa* [34], *B. juncea* [35], and *B. oleracea* [36]. In radish, although the upregulation of *RsTT8* expression has been associated with red skin or red flesh [33,37], polymorphisms in *RsTT8* associated with radish skin or flesh color have not yet been identified.

In this study, we isolated the *RsTT8* gene from four radish genotypes with different colored skin and flesh tissues and detected several polymorphisms (insertion/deletion mutations [InDels], single nucleotide polymorphisms [SNPs], and simple sequence repeats [SSRs]) between red-fleshed and white-fleshed accessions. The four radish genotypes showed no differences in *RsTT8* promoter activity; however, the expression level of *RsTT8* correlated with flesh color. Therefore, we developed and validated PCR-based molecular markers based on these polymorphisms to distinguish between radish accessions with different colored flesh. These markers could be efficiently applied to radish breeding for the development of red-fleshed radish cultivars.

## 2. Results

### 2.1. Selection of Radish Inbred Lines with Different Taproot Coloration

Radish inbred lines with different taproot skin and flesh colors (RsRf, GsRf, RsWf, and WsWf) were used in this study (Figure 1). The anthocyanin content of skin and flesh of each inbred line was measured just after the hypocotyl skin peeled off (ca. 35 days post-germination) (the skin of GsRf still exhibited anthocyanin accumulation) (Figure 1A). The coloration of the flesh was most distinct at the time of harvest, and the skin color of the RsRf inbred line used in this study was not as red as seen in other RsRf radishes.

### 2.2. RsTT8 as a Target Gene for Affecting Anthocyanin Accumulation in Radish

The presence of anthocyanin biosynthesis-related genes was determined by PCR (Figure 2A), and their expression in different organs or tissues was examined by semi-quantitative RT-PCR (Figure 2B). Even though the transcript level of EBGs, such as *RsCHS,* RsF3H, and *RsF3′H*, was highly expressed in flesh tissue of Rf compared to Wf. The expression of target genes of the MBW complex formed by RsTT8 (RsMYB and RsTTG1), such as *RsDFR* and *RsLDOX* genes, was consistent with anthocyanin accumulation.

Expression levels of *DFR, LDOX*, and *TT8* showed a strong correlation with anthocyanin accumulation in radishes. We selected the *RsTT8* gene, which encodes an important component of the MBW complex, for further analysis as a putative regulator of anthocyanin accumulation in radish.

### 2.3. Cloning and Sequence Analysis of RsTT8

The *RsTT8* gene was isolated from all four radish inbred lines using gene-specific primers (Appendix A), which were designed based on the sequence information available in various databases (see Materials and Methods). The sequence of at least 10 independent *RsTT8* clones per genotype was analyzed to identify putative paralogs. Primer positions and *RsTT8* gene structure are shown in Figure 3A. Nucleotide sequences of *RsTT8* amplified from the different inbred lines and their alignment are shown in Appendix A. On the basis of sequence information, a schematic of the genomic structure of *RsTT8* was redrawn to show several important residues or regions (Figure 3B). Several polymorphisms, such as SNPs, InDels, and SSRs, were identified between red-fleshed and white-fleshed radish lines (Appendix A, Table 1). An InDel immediately upstream of the transcription start site (TSS) and a dinucleotide (TC) SSR at the beginning of intron 2 were detected between red-fleshed and white-fleshed lines (Figure 3B). Phylogenetic analysis of *RsTT8* nucleotide sequences clearly distinguished the red-fleshed lines from white-fleshed lines (Figure 4), suggesting that radish germplasm used in this study are inbred lines (not doubled haploid lines), and *RsTT8*-Wf alleles are distinct from *RsTT8*-Rf alleles. However, polypeptides encoded by *RsTT8* genes amplified from all four inbred lines were well conserved (Figure 5), suggesting that the different anthocyanin accumulation levels among the four inbred lines are caused by differences in *RsTT8* expression levels.

### 2.4. Promoter Analysis of RsTT8

Regardless of the taproot color, all four radish inbred lines contained the *RsTT8* gene (Figure 2A). However, the expression of *RsTT8* was not detected in white tissues (Figure 2B). Although well-known *cis*-acting elements were conserved in *RsTT8* genes of all four inbred lines, various polymorphisms were detected in the promoter regions, which led us to speculate that promoter activity of *RsTT8* differs between red-fleshed and white-fleshed genotypes. However, the inbred lines showed no differences in *RsTT8* promoter activity (Figure 6), suggesting that radish taproot flesh color is determined by *RsTT8* expression level, which is regulated by an unknown upstream factor(s).

### 2.5. Development of Molecular Markers Associated with Flesh Color

Red-fleshed and white-fleshed accessions could be distinguished based on sequence polymorphisms in *RsTT8* (Figure 4). To develop molecular markers capable of distinguishing between red-fleshed and white-fleshed lines, we designed four PCR primer sets specific to different regions of the *RsTT8* gene: (1) InDel, present immediately upstream of the TSS; (2) dinucleotide (TC) SSR, present at the beginning of the 1st intron; (3) coding sequence (CDS), forward primer was designed using the InDel present in intron 5, and the reverse primer was designed using SNPs in exon 6; and (4) promoter, forward and reverse primers were designed using SNPs present between -979 and -818 bp (relative to the TSS). These four primer sets were tested on four inbred lines, three breeding materials, and four cultivars of radish (Figure 7). All four primer sets efficiently discriminated between red-fleshed and white-fleshed accessions. Radish cultivars RedSun, Rupr, and Anthopol, appeared to be heterozygous.

To further test the utility of these primer sets, they were used for genotyping 14 radish accessions obtained from Hankook Seed Co. and one radish accession purchased from a local market (Figure 8). All primer sets accurately identified most red-fleshed inbred and recombinant lines, although R-346 (RsRf) seemed to be heterozygous. Two white-fleshed radish accessions contained *RsTT8*-Wf alleles. However, Chengwoo, a white-fleshed accession, contained Rf-type SSR. We also applied the markers to newly-developed inbred lines (Figure 9). Eight red-fleshed inbred lines showed a perfect correlation between genotype and phenotype, but two inbred lines, which were still segregating flesh color into Green or Red, appeared to be ‘white flesh.’ When we examined three more individuals, one of the 2225 lines turned out ‘red flesh.’ These data indicate that red-fleshed lines could be identified from a collection of radish genotypes, and escaping from segregation of green/red flesh color would be very difficult.

Since the SSR marker genotype did not match the phenotype in a few samples (Figure 7 and Figure 8), we examined the SSR regions in 19 radish genotypes obtained from the NAC using the SSR-specific primer set. Sequence analysis revealed tremendous variation in the number of TC repeats between the red- and white-fleshed accessions and also among the white-fleshed genotypes (Appendix A). This result suggests that the SSR marker is not suitable for distinguishing radish lines with different colored flesh. Thus, InDels and SNPs identified in the *RsTT8* gene are ideal for the development of molecular markers for determining flesh coloration.

## 3. Discussion and Conclusions

Recently, several research groups performed transcriptomic analysis to identify the genes controlling anthocyanin accumulation in radish [3,33,37,38,39,40,41,42,43,44] and found that major genes and their regulatory transcription factors involved in the flavonoid biosynthetic pathway are essential for anthocyanin accumulation. However, none of these genes represents a novel regulatory gene that controls anthocyanin biosynthesis in a spatiotemporal or tissue-specific manner.

A few studies have investigated the role of *RsTT8* in anthocyanin biosynthesis in radishes. The altered expression of *TT8* due to TE insertion or a mutation decreases anthocyanin accumulation in *B. rapa* [34] and *B. juncea* [35]. In *B. juncea*, an insertion of 1279 bp in exon seven of the *TT8* gene disrupts its function, and the presence of a C→T SNP in exon seven results in the production of a truncated protein. On the other hand, upregulation of *RsTT8* is associated with red skin or red flesh color in radish [33,37]. Similar to the results of the current study, four SNPs found in the first three exons of *BoTT8* are associated with green or purple color in broccoli (*B. oleracea* var. *italica* L.) [36]. However, the presence of polymorphisms in *RsTT8* and their association with radish taproot coloration have not been reported to date.

In this study, we identified a large number of polymorphisms in the *RsTT8* gene isolated from red-fleshed and white-fleshed radish inbred lines and showed that some of these polymorphisms are associated with flesh color (Table 1, Figure 3 and Figure 5). Regardless of the flesh color, all radish genotypes contained the *RsTT8* gene (Figure 2A). Moreover, no TE insertion was found in *RsTT8* in any of the four inbred lines, suggesting that *RsTT8* was functional in all four lines. However, *RsTT8* did not express in tissues lacking anthocyanin accumulation, such as radish lines with white flesh or white skin color (Figure 2B), and the expression of *RsTT8* was not dependent on polymorphisms in the promoter (Figure 3 and Figure 6). Despite the importance of *TT8* in the regulation of anthocyanin biosynthesis, the mechanism underlying its regulation has not been identified yet. Nonetheless, the transcript level of *RsTT8* is correlated with those of *RsDFR* and *RsLDOX* (Figure 2B), indicating the expression of *RsTT8* is associated with that of two major LBGs.

Limited information is available on the genetics of flesh color in radish. Karyotype analysis suggests that the genetics underlying red flesh coloration in radish has evolved recently [45]. Several breeders speculate that anthocyanin accumulation is controlled in a recessive manner in radish (data not published). Epigenetic modifications of the genes encoding regulators of anthocyanin biosynthesis have been closely linked to color changes in crops, including radish. Changing the DNA methylation status of these regulatory genes affects the activities of the anthocyanin biosynthetic pathway. For example, methylation of the *PcMYB10* promoter in European pear (*Pyrus communis*) is associated with green-skinned spots in a red pear cultivar [46]. However, Wang et al. (2020) insist that radish flesh color is under epigenetic control; the authors propose that white-fleshed radish originated from a red-fleshed accession following the hypermethylation of *RsMYB1* by a CACTA transposon, resulting in considerable downregulation of *RsMYB1* expression [47]. However, the above type of repression did not occur (Figure 2B) in the radish accessions used in this study, indicating that epigenetic modifications are not an absolute determinant of flesh color. However, the F_10_ inbred lines 2224 and 2225 (Figure 9) should be examined to determine whether the epigenetic regulation of *RsMYB1* affects flesh color in radish to confirm regulation of anthocyanin biosynthesis in terms of epigenetic control with further study.

Marker development and application are very important for breeding radish cultivars because the selection of genotypes for taproot coloration is impossible at the seedling stage. The outer layer of the hypocotyl in radish does not reflect anthocyanin accumulation in mature tissues because this layer peels off before maturation. The skin color of a mature taproot is therefore different from that of a young seedling. This is quite different from *Brassica* seedlings, in which the seedling stage is adequate for the selection of anthocyanin-accumulating genotypes. Therefore, DNA-based molecular markers are highly valuable for radish breeding.

## 4. Materials and Methods

### 4.1. Plant Materials

Four radish inbred lines were used in this study: HKR-397 (GsRf), HKR-275 (RsRf), HKR-519 (WsWf), and HKR-513 (RsWf). Seeds of these inbred lines were kindly supplied by Hankool Seed Co. Ltd. (143-71 Yucheon-Ro, Pyeongtaek-si, Gyeongii-Do, Korea). Additional seeds or tissues used in this study were provided by Hankook Seed Co. or purchased from the local market. Inbred lines (2217 to 2226) used for marker development were bred from the 4th to the 11th filial generation. Radish germplasm used for determining the number of SSRs was obtained from the National Agrobiodiversity Center (NAC), Jeonju 54874, Republic of Korea.

Seeds were sown in pots with multiple perforations at the bottom. After germination, seedlings were transplanted into appropriate pots and grown in either an environmentally controlled growth chamber or in a glass greenhouse at Chungnam National University, Daejeon, Korea. Plants were photographed prior to sampling at four distinct stages (young seedling stage, before and after the peeling off of hypocotyl skin, and harvest stage). The sampled tissues were frozen until needed for DNA extraction.

### 4.2. Genomic DNA Extraction and Cloning and Sequence Analysis of RsTT8

Genomic DNA was isolated from the leaf tissues of all four radish inbred lines using the DNeasy Plant Mini Kit (QIAGEN, Hilden, Germany). To amplify the *RsTT8* gene, primers were designed on the basis of sequence information available at the National Center for Biotechnology Information (NCBI) and two radish genome databases (http://radish-genome.org/ and http://www.nodai-genome-d.org/, accessed on 9 March 2019) (Appendix A). PCR was performed under the following conditions: initial denaturation at 94 °C for 5 min, followed by 30 cycles of amplification (94 °C for 30 s, 58 °C for 30 s, and 72 °C for 2–5 min), and a final extension at 72 °C for 7 min. PCR products were purified using the LaboPass Gel Extraction Kit (Cosmogenetech, Seoul, Korea) and cloned into the TA vector using the T&A Cloning Kit (RBC Bioscience Co., Taipei, Taiwan). The resulting plasmids were transformed into *Escherichia coli* (DH5α) cells. Plasmid DNA was purified using DNA-Spin (Intron Biotech. Inc., Seongnam, Korea), and sequenced. To obtain all possible *RsTT8* sequences present in each inbred line, at least 10 clones of each line were sequenced and analyzed. Any possible PCR and/or sequencing errors were eliminated by aligning independent sequences. The *RsTT8* gene sequences amplified from all four inbred lines were deposited in the NCBI GenBank database under the accession numbers: RsWf_TT8_1 (MW657230), RsWf_TT8_2 (MW657231), GsRf_TT8_1 (MW657232), GsRf_TT8_2 (MW657233), RsRf_TT8_1 (MW657234), RsRf_TT8_2 (MW657235), WsWf_TT8_1 (MW657236), and WsWf_TT8_2 (MW657237).

PCR was conducted to determine the presence or absence of genes in radish samples using 5 ng of genomic DNA under the following conditions: initial denaturation at 94 °C for 5 min, followed by 29 cycles of amplification (94 °C for 30 s, 60 °C for 30 s, and 72 °C for 30 s), and a final extension at 72 °C for 5 min. The resulting PCR products were separated by electrophoresis on 1.5% agarose gel.

### 4.3. RNA Extraction and Gene Expression Analysis

Total RNA was extracted from plant samples using the RNeasy Mini Kit (Qiagen, Hilden, Germany) and then treated with RNase-free DNase (Promega, Madison, WI, USA) to remove genomic DNA contamination. Then, cDNA was synthesized by semi-quantitative reverse transcription PCR (RT-PCR) using the Avian Myeloblastosis Virus (AMV) One-step RT-PCR Kit (Takara, Kyoto, Japan) and gene-specific primers (Appendix A) under the following conditions: initial denaturation at 94 °C for 5 min, followed by 27 cycles of amplification (94 °C for 30 s, 60 °C for 30 s, and 72 °C for 30 s), and a final extension at 72 °C for 7 min. PCR products were separated on 1.5% agarose gels and stained with ethidium bromide.

### 4.4. Anthocyanin Extraction and Quantification

Anthocyanin content was measured as described previously [48]. Briefly, 1 mL of 1% (*v*/*v*) HCl-methanol was added to 0.3 g of the powdered leaf tissue. The sample was shaken gently in the dark at room temperature for 24 h and then centrifuged at 13,000× *g*. The upper aqueous phase was collected, and its absorbance was measured at 530, 620, and 650 nm. Relative anthocyanin content (RAC) was calculated according to the following equation:(1)$$\mathrm{RAC}=\left(A_{530}-A_{620}\right)-0.1\times \left(A_{650}-A_{620}\right)$$
where A_530_, A_620_, and A_650_ represent the absorbance of the aqueous layer at 530, 620, and 650 nm, respectively.

Total anthocyanin content (TAC; mg anthocyanin/100 g fresh weight) was then calculated on the basis of the reference, cyanidin-3-glucoside [49], according to the following equation:(2)TAC=RAC×MW×DF×1000×∋
where MW is the molecular weight of cyanidin-3-glucoside (449.2 g/mol), DF is the dilution factor and, ϶ is the cuvette optical path length (1 cm).

### 4.5. Promoter Activity Assay

To analyze the *RsTT8* promoter activity, the approximately 870-bp sequence upstream of the TSS of the *RsTT8* gene was amplified from the genomic DNA of WsWf and RsRf inbred lines using sequence-specific primers (Appendix A). The amplified fragments were inserted into the T&A cloning vector (Real Biotech Co., Taipei, Taiwan), and the presence of the insert was confirmed by sequencing. The resulting recombinant plasmids were digested with *Bam*HI and *Nco*I, and the released fragments were subcloned into the pCambina3301-GUS binary vector digested with the same enzymes. The resulting constructs were introduced into *Arabidopsis thaliana* ecotype Columbia (Col-0; wild type) plants by *Agrobacterium*-mediated transformation using the floral dip method [50]. The transformed plants were selected using 0.1% Basta herbicide and confirmed by PCR. Homozygous T_3_ lines containing a single copy of the T-DNA, determined by 3:1 (Basta resistant: sensitive) segregation ratio, were selected for the analysis of promoter activity.

Wild-type and transgenic *Arabidopsis* plants were grown in a growth chamber at 22 °C under a 16-h light/8-h dark photoperiod and 100 μmol m^−2^s^−1^ light intensity. To analyze *RsTT8* promoter activity, wild-type and transgenic seeds were surface-sterilized with 30% bleach containing 0.1% Triton X-100, cold-stratified at 4 °C for 3 days, and then sown on solid half-strength Murashige and Skoog (1/2 MS) medium supplemented with or without 90 mM sucrose. Three plants of each genotype were sampled for analyzing *GUS* expression by RT-PCR, and another three plants per genotype were used to perform GUS staining. To perform GUS staining, plants were incubated in GUS-staining solution (1 mM X-GlucA in 100 mM sodium phosphate [pH 7.0] containing 5 mM Na_2_EDTA, 0.5 mM potassium ferrocyanide, 0.5 mM potassium ferricyanide, and 0.1% Triton X-100) for 16 h at 37 °C. After staining, the plants were washed with 95% ethanol at room temperature until the wild-type plants appeared clear.

### 4.6. Marker Development and Validation

DNA was extracted from leaf samples of various radish genotypes using the DNeasy Plant Mini Kit (QIAGEN, Hilden, Germany). PCR was performed in a 20 μL volume containing 5–20 ng genomic DNA template, 10 pmol each of forward and reverse InDel-, SNP-, or SSR-specific primers (Appendix A), and 4 µL of 5X PCR Master Mix (Elpisbiotech, Daejeon, Korea), under the following conditions: 94 °C for 5 min, followed by 30 cycles at 94 °C for 30 s, 62 °C for 30 s, and 72 °C for 30 s, with a final extension at 72 °C for 5 min. To verify the presence of genomic DNA, *B. oleracea Actin 1* (*BoActin1*) was simultaneously amplified. The PCR products were separated by electrophoresis on 1% agarose gels. Additionally, DNA sequences flanking the InDels and SSRs were amplified by PCR under the same conditions as those used for marker validation (Appendix A). The PCR products were sequenced, and their association with radish flesh coloration was examined.

## Figures and Tables

**Figure 1 plants-10-01386-f001:**
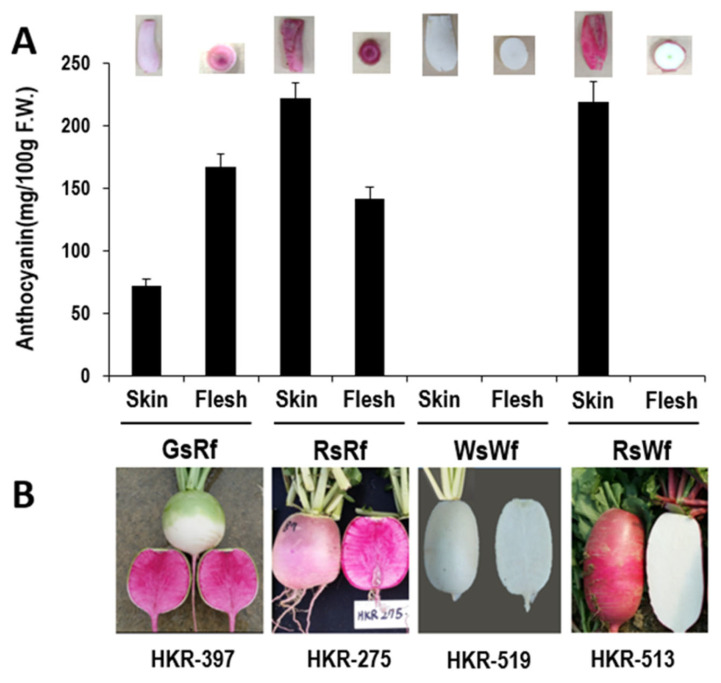
Four radish samples used in this study: green skin/red flesh (GsRf), red skin/red flesh (RsRf), white skin/white flesh (WsWf), and red skin/white flesh (RsWf) with inbred number (bottom). (**A**) Anthocyanin contents measured with 35-day-old samples. Real photos are on top. (**B**) Photographs taken at the harvest period.

**Figure 2 plants-10-01386-f002:**
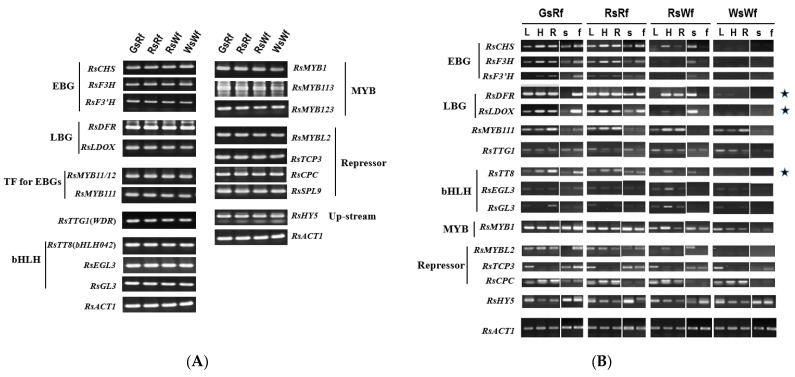
Presence (**A**) and expression (**B**) of genes related to anthocyanin accumulation in four radishes. (**A**) All genes were amplified by genomic DNA PCR. (**B**) Expression levels of genes. L, leaves, H, hypocotyls, R, roots, s, skin, and f, flesh. Asterisks indicated that expression levels were tightly related to anthocyanin accumulation.

**Figure 3 plants-10-01386-f003:**
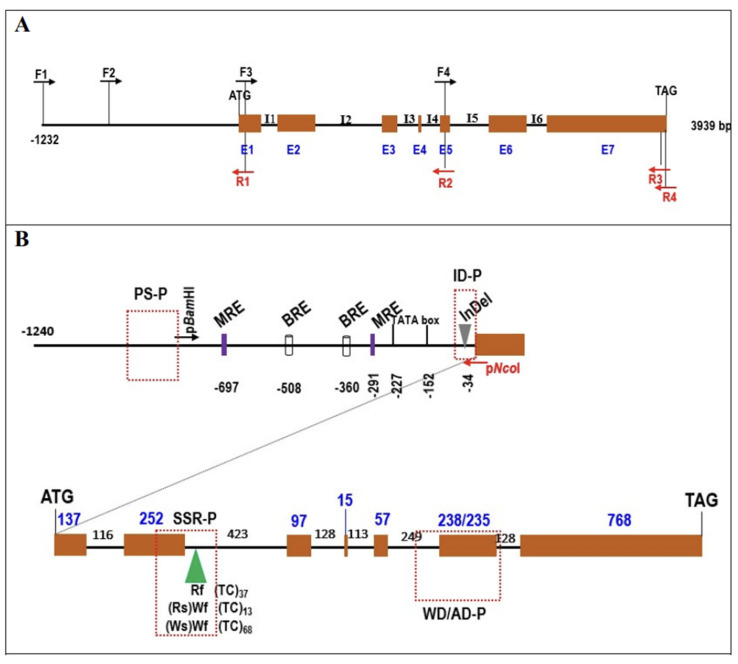
Schematic representation of genomic DNA of *RsTT8* genes with primer position for cloning (**A**) and genomic organization of cloned *RsTT8* genes (**B**). (**A**) Exon and intron are represented by box and solid line, respectively. I and E indicate intron and exon, respectively. Arrows indicated primer position and direction with forward (F) and reverse (R). (**B**) Genomic organization of cloned *RsTT8* genes with promoter regions and 7 exons (box) and 6 introns (solid line). Numbers indicated nucleotide sequence (bp), and InDel and SSR regions were also marked. MRE, MYB-recognition motif; BRE. bHLH-recognition motif. Arrows named p*Bam*HI and p*Nco*I indicated primers for promoter::GUS construction. Four boxes of dotted red lines indicate regions used in marker validation: PS-P, region amplified by promoter-specific primer set (−979~−818) (Rf-P/Wf-P in Appendix A); ID-P, region amplified by 14 bp InDel-discriminating primer set; SSR-P, region amplified by TC SSR including primer set; WD/AD-P, region amplified by primer set that discriminated one CAT InDel and three SNPs (Rf/Wf-P in Appendix A).

**Figure 4 plants-10-01386-f004:**
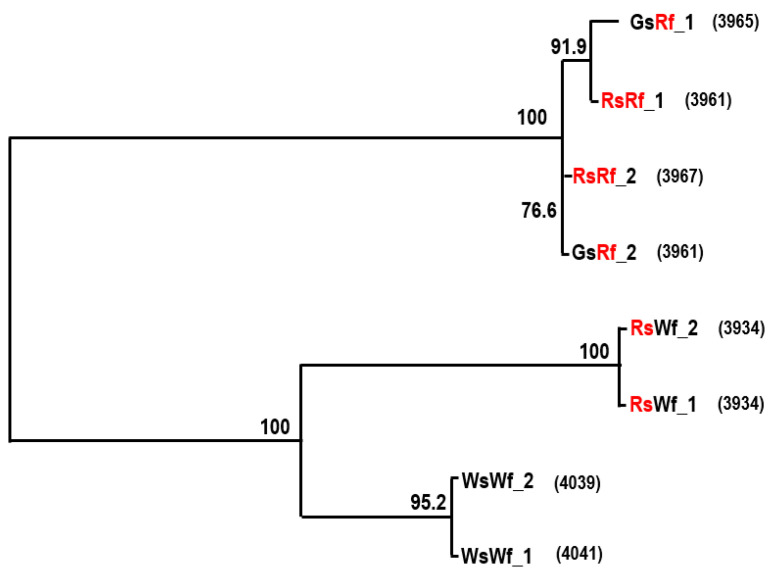
Phylogenetic relationship among *RsTT8* alleles. Numbers in parenthesis indicate nucleotides of each allele.

**Figure 5 plants-10-01386-f005:**
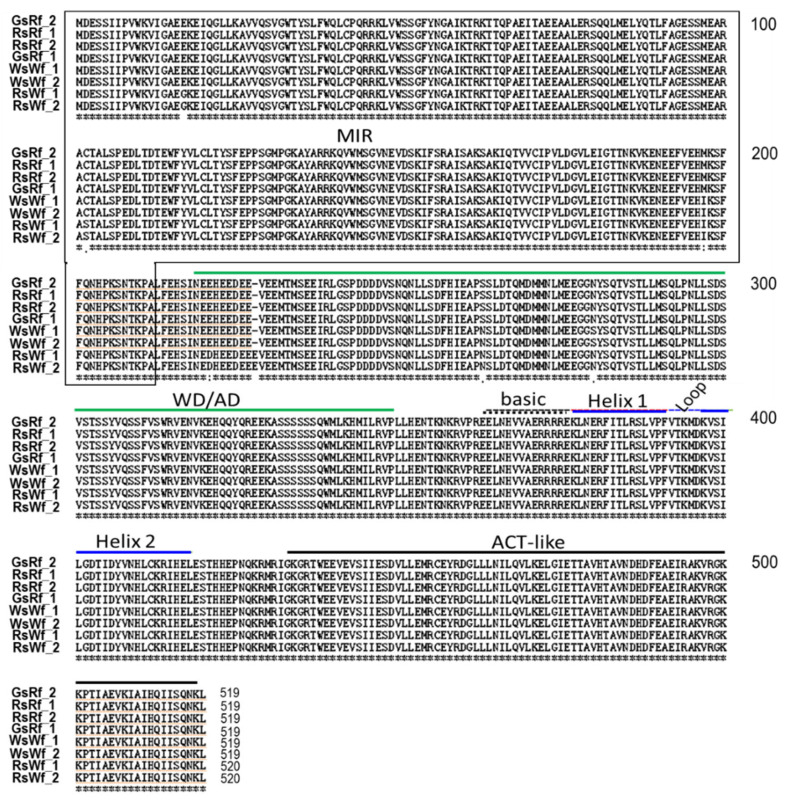
Alignment of amino acids of RsTT8 from four radishes used in this study. MIR (MYB-interacting domain), WD/AD (WDR interacting region through the acidic domain), bHLH region, and ACT-like domain were well conserved among RsTT8s, indicating that resulting proteins keep their activity. Regulation may depend on the expression level. Numbers on the right denoted the number of amino acid residues.

**Figure 6 plants-10-01386-f006:**
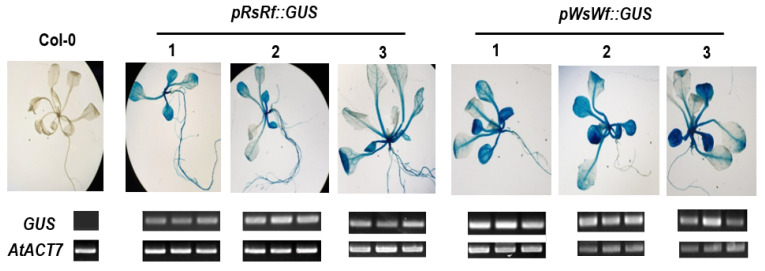
Promoter activity assay by GUS stain and RT-PCR.

**Figure 7 plants-10-01386-f007:**
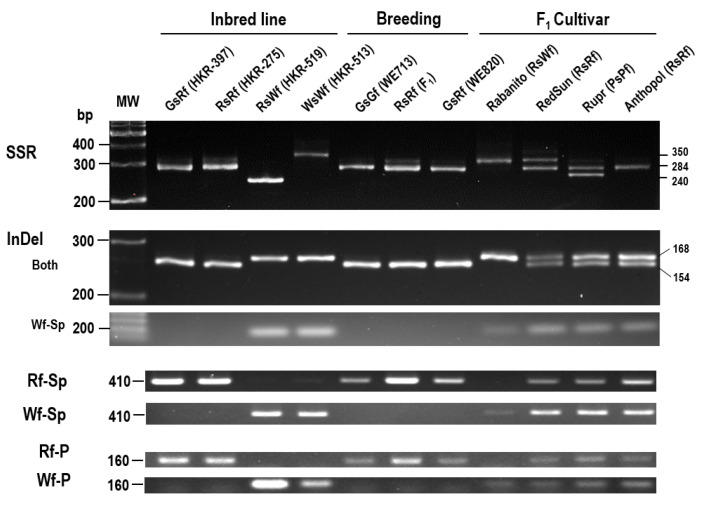
Development of PCR-markers derived from polymorphisms (SSR, InDel, and SNPs). (TC)-SSRs were present in the 2nd intron ranging from (TC)_13_ to (TC)_68_. InDel was 14 nucleotide insertion in Wf gene and present at the 35th nucleotide of ATG upstream. Rf-Sp and Wf-Sp indicated that Rf-specific and Wf-specific amplification, respectively, which was designed using CAT InDel in 5th intron and SNP in exon 6. Rf-P/Wf-P indicated SNPs presented in the promoter regions between −979 to −818 nucleotide from ATG start codon. Primer sequences were listed in Appendix A. Numbers on the right denoted size of band (bp).

**Figure 8 plants-10-01386-f008:**
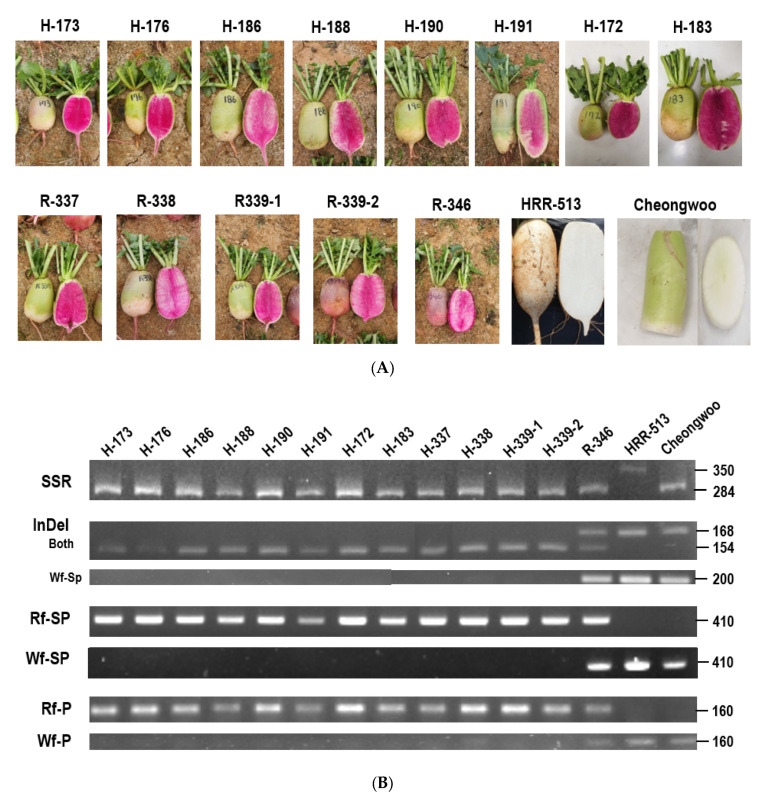
Validation of PCR-markers using F_1_ cultivars in Korea. H and R indicate inbred and recombinant lines, respectively, from Hankook Seed Co. Numbers on the right, denote size of band (bp). (**A**) Morphology of taproots used in this study. (**B**) Results of PCR amplification.

**Figure 9 plants-10-01386-f009:**
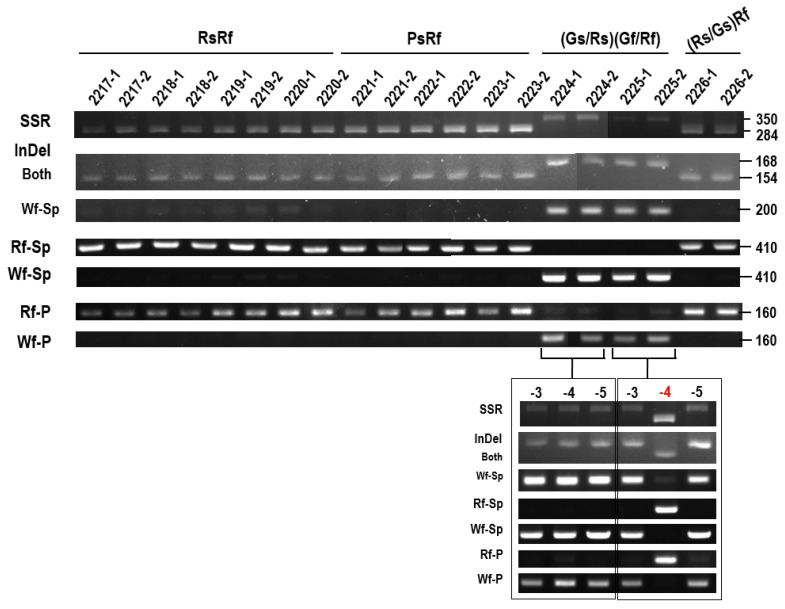
Validation of PCR-markers using newly developing radish lines in Sejong University. -P indicates the pink color and that lines were still segregated in terms of taproot color. Numbers on the right denoted size of band (bp). 2219 and 2221, F_4_ generation; 2220, F_5_ generation; 2217 and 2218, F_6_ generation; 2223–2226, F_10_ generation; 2222, F_11_ generation.

**Table 1 plants-10-01386-t001:** List of polymorphisms found in *RsTT8* genes.

Region	Between Rf and Wf			WsWf-Specific			RsWf-Specific		
	SNP	InDel	SSR	SNP	InDel	SSR	SNP	InDel	SSR
Promoter	16	2	1	2	4	1	2	-	-
Exon	2	-	-	1	-	-	3	1	-
Intron	10	1	2	3	3	2	4	-	1
Total	28	3	3	6	7	3	9	1	1

## Data Availability

Not applicable.

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
