# Peer review of "Development of Molecular Markers for Predicting Radish (Raphanus sativus) Flesh Color Based on Polymorphisms in the RsTT8 Gene"

_plants, 2021, doi:10.3390/plants10071386_

Round 1
Reviewer 1 Report
In the manuscript entitle: “Development of molecular markers for predicting radish (Raphanus sativus) flesh color based on polymorphism in the RsTT8 gene”, sequence analysis of RsTT8 on 4 different phenotype radish lines was performed with PCR and RT-PCR technique related to amplification of anthocyanin accumulation gene family. Molecular markers is developed from sequence analysis and prove efficient feature in discrimination of red-fleshed and white- fleshed radish, serving for red radish breeding. However, there are several major concern in this manuscript which need to be addressed to support conclusion from the study
-
The tittle of manuscript only about radish flesh color related to anthocyanin accumulation gene, whether using radish inbred lines with different taproot skin is necessary for analysis? (Figure 1). Beside, in the development of marker in 2.5 section, the author only focused on the discriminate red and white-fleshed radish lines.
-
The explanation for figure 2B in line 115 and 116 are quite simple to emphasis strong correlation of DFR, LDOX and TT8 genes compare with other anthocyanin synthesis gene subfamily. The detail explanation is needed.
-
In line 116-118, the author mentioned that the results showed strong expression level of DFR, LDOX and TT8 with anthocyanin accumulation. What is feature make RsTT8 become selection for target gene compare to remaining gene subfamily? The author should emphasis to make logical characteristic for subsequent analysis.
-
The author mentioned real time PCR in M&M, however, this reviewer could not find that result. Only reverse transcription PCR shown in manuscript.
Author Response
Dear Editor:
We would like to thank you for the opportunity of revising our manuscript. We have addressed a points raised by the reviewer 1 in the text with red color.
Comments and Suggestions for Authors and responses
Reviewer 1:
In the manuscript entitle: “Development of molecular markers for predicting radish (Raphanus sativus) flesh color based on polymorphism in the RsTT8 gene”, sequence analysis of RsTT8 on 4 different phenotype radish lines was performed with PCR and RT-PCR technique related to amplification of anthocyanin accumulation gene family. Molecular markers is developed from sequence analysis and prove efficient feature in discrimination of red-fleshed and white- fleshed radish, serving for red radish breeding. However, there are several major concern in this manuscript which need to be addressed to support conclusion from the study.
Point 1: The tittle of manuscript only about radish flesh color related to anthocyanin accumulation gene, whether using radish inbred lines with different taproot skin is necessary for analysis? (Figure 1). Beside, in the development of marker in 2.5 section, the author only focused on the discriminate red and white-fleshed radish lines.
Response 1: Most part of taproot radish is occupied by flesh tissue, which is important region for color (anthocyanin accumulation). Therefore, we have focused on flesh color for our research and fortunately we have found polymorphisms in RsTT8 gene discriminating flesh color.
Point 2: The explanation for figure 2B in line 115 and 116 are quite simple to emphasis strong correlation of DFR, LDOX and TT8 genes compare with other anthocyanin synthesis gene subfamily. The detail explanation is needed.
Response 2: Even though the expression of EBGs, such as RsCHS, RsF3H, and RsF3’H, was high in flesh tissue of Rf tissues compared to Wf tissue, but the expression of target gens of MBW complex formed by RsTT8 (RsMYB and RsTTG1), such as RsDFR and RsLDOX genes, was consistent with anthocyanin accumulation.
Point 3: In line 116-118, the author mentioned that the results showed strong expression level of DFR, LDOX and TT8 with anthocyanin accumulation. What is feature make RsTT8 become selection for target gene compare to remaining gene subfamily? The author should emphasis to make logical characteristic for subsequent analysis.
Response 3: As we have mentioned in the text, RsTT8 is important component of MBW complex which regulates LBD gene expression, DFR and LDOX, thereby controlling anthocyanin bionsynthesis.
Point 4: The author mentioned real time PCR in M&M, however, this reviewer could not find that result. Only reverse transcription PCR shown in manuscript.
Response 4: We are really sorry about it. We eliminated qRT-PCR part from M&M.
Point 5: If one of the referees has suggested that your manuscript should undergo extensive English revisions, please address this issue during revision. We propose that you use one of the editing services listed at https://www.mdpi.com/authors/english or have your manuscript checked by a
native English-speaking colleague.
Response 5: We already done extensive English revisions with special edit company and reviewer is not indicated specifically. Therefore, it is very difficult to find any specific parts. Also, we tried to do best for writing manuscript. Please understand our situation at this moment.

Reviewer 2 Report
This manuscript is very well written and organized. All the necessary information is in the main document.
Introduction section describes the essential state of the art. Metods are adequate for the proposed search and conclusions are nicelly experimentally proved.
The authors detected new polymorphisms in a less explored gene that revealed very useful markers for breeding activities.
Therefore I recommend this manuscript for publication.
Although using
Author Response
Comments and Suggestions for Authors and response
This manuscript is very well written and organized. All the necessary information is in the main document. Introduction section describes the essential state of the art. Metods are adequate for the proposed search and conclusions are nicely experimentally proved. The authors detected new polymorphisms in a less explored gene that revealed very useful markers for breeding activities.
Therefore, I recommend this manuscript for publication.
Response 1: Not applicable

Round 2
Reviewer 1 Report
The authors addressed all my comments.
Author Response
Dear Reviewer:
I am sorry to bother you. In first round, I already addressed reviewer's coments and suggestions.
At this moment, How and what do I need to do for minor revision?
Just in case, I upload final revised version. If I do not understand something, please let me know as soon as possible.
Thank you for your kind help.
On behalf of all the authors,
Sincerely yours,
Man-Ho Oh
Professor and corresponding author
